# Three-Dimensional Point Cloud Data Pre-Processing for the Multi-Source Information Fusion in Aircraft Assembly

Rupeng Li [1], Weiping He [1] and Siren Liu [2],*

[1] Cyber-Physical Interaction Lab, Northwestern Polytechnical University, 127 West Youyi Road, Xi'an 710072, China
[2] Institute of Aeronautical Manufacturing Technology, COMAC Shanghai Aircraft Manufacturing Co., Ltd., Shanghai 200120, China
* Correspondence: liusiren@comac.cc

**Abstract:** Wing-body assembly is a key part of aircraft manufacturing, and during the process of wing assembly, the 3D point cloud data of the components are an important basis for attitude adjustment. The large amount of measured point cloud data and the obvious noise affect the quality and efficiency of the final assembly. To address this problem, research on the pre-processing method of the component point cloud data is carried out. Firstly, a feature-enhanced point cloud resampling method is proposed to preserve key features such as part contours in the resampling process. Then, a multi-scale point cloud data noise filtering method is proposed, which can effectively filter out the outliers. The experimental results show that the proposed method improves the speed and accuracy of the subsequent point cloud analysis effectively and is successfully applied to the assembly process of a large passenger aircraft, laying the foundation for high-quality assembly.

**Keywords:** aircraft assembly; point cloud resampling; noise filtering

## 1. Introduction

Aircraft assembly is a key part of aircraft manufacturing, and it directly determines the efficiency and quality of manufacturing [1,2]. In the process of aircraft assembly, wing-body assembly is regarded as the most important assembly process because the wing box is the main load-bearing structure of the aircraft [3–5]. In the wing-body assembly process, the wing is fixed on the positioner and the fit of the outer wing and the central wing box is achieved by the adjustment of the wing position and attitude, and the assembly quality is ensured by the assembly gap between the wing and the fuselage [6]. Due to gravity, positioner load, time-varying temperature, and other factors, the composition docking surface profile of the assembly process is time-varying, and wing-body assembly can only be completed through the iterative process of profile measurement and pose adjustment [7]. In this process, the 3D point cloud data of the joint surface are often obtained by digital measurement instruments such as laser scanners [5,8] and LIDAR [9,10] and fed to the positional control system. The obtained point cloud data, however, are often huge and contain noisy data due to the huge volume of aircraft components and interference from other equipment, which reduces the analysis and processing speed of measurement data and increases the difficulty of data storage [8,11]. The processing and fusion of point cloud data is the basic technology for obtaining high-quality information in the assembly process, which can effectively reduce the interference brought by the environment and improve the analysis efficiency, and ultimately help to improve the quality and efficiency of aircraft assembly.

The pre-processing of point cloud data is the basis for subsequent point cloud alignment, gap evaluation and other aspects [12–14]. The purpose of pre-processing is to reduce the amount of data and effectively filter the noise while maintaining the key features of the components as much as possible [15–17]. For the pre-processing problem of point cloud

data, the two main dimensions of point cloud resampling and denoising are carried out at this stage. Jovancevic [18] et al. used the moving least squares method to resample and smooth point cloud data for the problem of point cloud measurement errors and outliers, and the related method was applied to A320 surface defect detection. Chen et al. [19] proposed a resampling strategy based on graph filter to address the difficulty of storing and processing large-scale point cloud data. Paoli et al. [20] improved the speed of the 3D scanning of hull surfaces by removing outliers through statistical filters and performing curvature-aware resampling by removing redundant information in low-curvature regions. Dai et al. [21] used tensor voting theory to process point cloud data to infer hidden structure information from a 3D point cloud with strong noise and outliers and verified the effectiveness of the proposed method in aircraft skin seam detection. Ning et al. [22] proposed a noise point removal method based on geometric feature constraints for the point cloud noise problem caused by occlusion, which achieves the effective removal of sparse and isolated outliers. However, the above methods are mainly for small-range point cloud data, and are less efficient and less adaptable for large-scale point cloud data. In addition, the point cloud data of aircraft assembly is more diverse and complex, and the traditional methods cannot achieve a better noise removal effect.

This paper proposes a point cloud pre-processing method for the aircraft assembly process wing-body alignment problem. Specifically, a point cloud resampling method with feature enhancement is proposed to address the problem of the high cost of alignment time due to redundant point cloud data, and by combining nonuniform voxel mesh resampling and graph signal processing methods, it transforms the point cloud resampling problem into an optimization problem and preserves key information such as part profile as much as possible in the process of resampling. Secondly, a multi-scale point cloud data noise filtering method is proposed, which uses a median denoising method based on region segmentation to remove large-scale noise and an improved bilateral filtering method to remove small-scale noise. The proposed method can effectively ensure the efficiency and reliability of subsequent point cloud data processing and analysis.

## 2. Feature Enhancement Point Cloud Data Resampling

Aircraft components are generally large, and the data points can reach millions when the point cloud is scanned using measurement instruments such as scanners. The analysis of large point clouds is time-consuming, especially during the alignment process. In addition, the large point cloud also increases the burden of data storage. Therefore, the measured point cloud must be simplified to some extent. However, the simplification of the point cloud will undoubtedly affect the accuracy of the subsequent alignment. Traditional point cloud processing algorithms only consider the uniformity of sampling, such as the uniform resampling method with the help of voxel squares, but these methods cannot maintain the feature information of the original point cloud, which will eventually lead to worse alignment accuracy.

To address this problem, we propose a feature enhancement method for point cloud resampling, so that the simplified point cloud can have uniformity while maintaining features such as edges. Specifically, we design a resampling strategy that consists of two steps. First, the point cloud is resampled using voxel mesh combined with geometric sampling to simplify the point cloud. Next, the point cloud is further simplified by transforming the point cloud resampling problem into an optimization problem while preserving the point cloud features.

### 2.1. Nonuniform Voxel Mesh Resampling

The voxel mesh resampling, similar to lattice point sampling, is a grid of point clouds, with $(X_{\min}, X_{\max}), (Y_{\min}, Y_{\max}), (Z_{\min}, Z_{\max})$ as the range of values for the three-

dimensional directions of the point cloud, forming a rectangular enclosing box on the space, whose length, width, and height are:

$$L = X_{\max} - X_{\min} W = Y_{\max} - Y_{\min} H = Z_{\max} - Z_{\min} \tag{1}$$

In the 3D space, the mesh can be divided by unit length, which can eventually be divided into $N$ voxel spaces, which in turn can be replaced by the centers of all points in the voxel space.

Although the voxel mesh resampling method ensures that the point cloud shape remains unchanged, the feature information is easily lost at some corners where the curvature changes significantly, so the geometric sampling method is introduced to reduce the number of point clouds where the local curvature changes drastically. Geometric features can be used based on the Difference of Normal (DoN) method to first perform point cloud segmentation, segmenting two parts of the point cloud with obvious curvature change and gentle curvature change, and taking different sizes of voxels' down-sampling for the two-point clouds, respectively. For the point cloud with obvious curvature change, a smaller sized grid is used to retain more points. For the point cloud with insignificant curvature variation, a larger sized grid is used to reduce the number of point clouds and improve the computational efficiency.

Suppose the radii of two different neighborhoods of point $p$ are $r_l$ and $r_s(r_l > r_s)$, then the difference between the normal is calculated as shown in Equation (2):

$$\Delta\hat{n}(p, r_s, r_i) = \frac{\hat{n}(p, r_s) - \hat{n}(p, r_i)}{2} \tag{2}$$

where $\hat{n}(p, r_i)$ and $\hat{n}(p, r_s)$ are the normal vectors calculated from the radii.

The specific process is as follows.

(1) The DoN operator is calculated for the point cloud after de-contextualization, which is divided into two parts: those with drastic curvature change and those with gentle curvature change.

(2) Different sizes of resampling are used. For the part with large curvature change, a smaller size voxel mesh is used for resampling to retain the detail information to the maximum extent, and for the part with gentle curvature change, a larger size voxel mesh is used for resampling to minimize the number of plane points, and the number of points in the two parts after sampling is summed.

(3) Compare the sum of the two parts of the point cloud with the sampling space; if it is larger than the threshold value, the sampling size is adjusted to a larger size for the part with obvious curvature change and recalculated; if it is smaller than the threshold value, the sampling size is adjusted to a smaller size and the area with gentle curvature change is recalculated until the value range is satisfied.

*2.2. Secondary Resampling of Feature Enhancement*

To better characterize the point cloud resampling method, the resampling descriptor $\Psi$ can be defined, thus linearly mapping the original high-dimensional point set to the low-dimensional point set:

$$\Psi_{i,j} = \begin{cases} 1, & j = \mathcal{H}_i \\ 0, & \text{other} \end{cases} \tag{3}$$

where $\mathcal{H} = (\mathcal{H}_1, \ldots, \mathcal{H}_M)$ denotes resampling data labels so that the point cloud can be resampled from $N$-dimensions to $M$-dimensions by $\Psi X$.

The aerospace component point cloud is complex and dense, and to simplify the point cloud while maintaining its key features, we use the theoretical framework of graph signal processing (GSP), thus extending the relevant tools from classical discrete signal processing

to high-dimensional, complex point cloud data. Specifically, we use a Laplace filter to enhance the edge features in the point cloud.

$$W_{i,j} = \begin{cases} \exp\left(-\frac{\|\mathbf{x}_i - \mathbf{x}_j\|_2^2}{\sigma^2}\right) & \|\mathbf{x}_i - \mathbf{x}_j\|_2^2 \leq t \\ 0 & \text{Other} \end{cases} \tag{4}$$

where $W_{i,j}$ refers to the adjacency matrix. $\mathbf{x}_i$ and $\mathbf{x}_j$ represent the point cloud feature points. When the distance between two points in the point cloud is less than the threshold $t$, that is, the two points are connected, and the connection weights are calculated. Thus, the point cloud connectivity graph is constructed from the original point cloud data. Difference metrics of data points in point cloud $\widetilde{\mathbf{X}}$:

$$\widetilde{\mathbf{X}}_i = \mathbf{x}_i - \sum_j \widetilde{W}_{i,j} \mathbf{x}_j \tag{5}$$

$$\widetilde{W}_{i,j} = \frac{W_{i,j}}{\sum_j W_{i,j}} \tag{6}$$

It is easy to see from the above equation that if $\mathbf{x}_i$ is significantly different from its neighbors, then $\left\|\widetilde{\mathbf{X}}_i\right\|_2$ is greater, i.e., the data point is more representative of features such as edges in the point cloud. Thus, the feature loss during point cloud resampling can be defined by the following equation.

$$L_f(\mathbf{\Psi}) = \left\|\mathbf{\Psi}\widetilde{\mathbf{X}} - \widetilde{\mathbf{X}}\right\|_2^2 \tag{7}$$

However, if only point cloud features are retained, the set of nonfeatured points will be lost in large numbers, i.e., the point cloud after resampling will be very sparse, which is also not conducive to the subsequent point cloud alignment. Therefore, a trade-off must be made between feature retention and sparsity of the point cloud. To take into account the sparsity of the point cloud, we define the binarization matrix $B$ if $\mathbf{x}_j$ is within the neighborhood of $\mathbf{x}_i$, then $B_{i,j} = 1$. Furthermore, the sparsity loss can be defined by the following equation.

$$L_d(\mathbf{\Psi}) = \|B\mathbf{\Psi} - \alpha k\|_2^2 \tag{8}$$

where $\alpha$ and $k$ are hyperparameters that can be set as required, after which the point cloud resampling problem can be transformed into the following optimization problem:

$$\begin{aligned} \min_{\mathbf{\Psi}} & \|\mathbf{\Psi}\widetilde{X} - \widetilde{X}\|_2^2 + \lambda \|\mathbf{A}\mathbf{\Psi} - \alpha k\|_2^2 \\ s.t. \quad & \mathbf{\Psi}_{i,i} \in \{0,1\}, i = 1, 2, \ldots, N \\ & \mathbf{\Psi}_{i,j} = 0, i \neq j \\ & \text{tr}(\mathbf{\Psi}) = \alpha N \end{aligned} \tag{9}$$

The problem can be solved efficiently by methods such as the interior point method after simplification.

## 3. Multi-Scale Noise Filtering Method

For the point cloud noise problem, the robust principal component analysis method is first used to transform the coordinate system, then the median denoising method based on region splitting is used to denoise the large-scale noise, and finally the improved fast bilateral filtering method is used to denoise the small-scale noise (Figure 1).

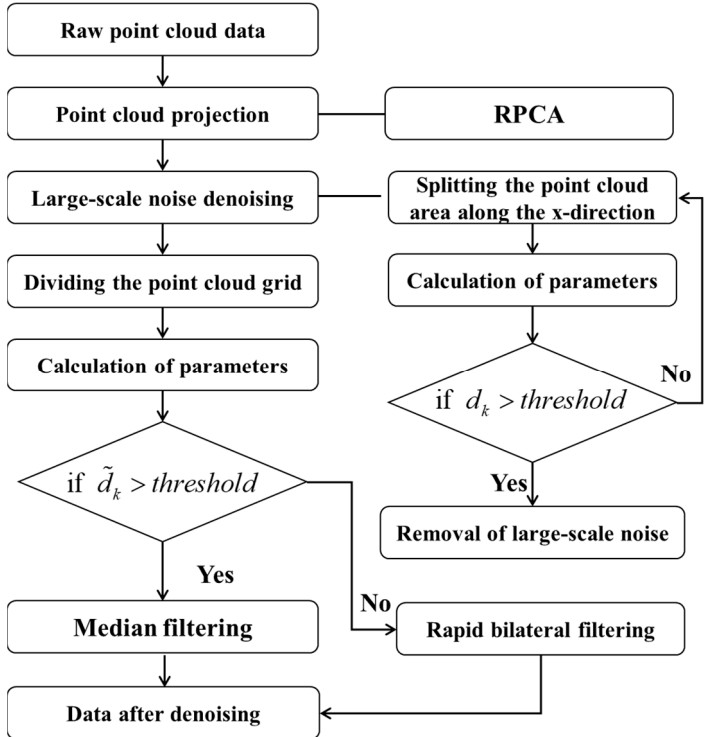

**Figure 1.** Denoising algorithm flow.

*3.1. Coordinate System Conversion Based on Principal Component Analysis*

The aerial component point cloud data are mainly collected through the scanner, and the projection direction of the maximum data volume of point cloud data cannot be obtained directly. Considering the problem of point cloud data noise, we first solve the solution direction of the maximum data volume of point cloud data by robust principal component analysis [23] (RPCA), i.e., we obtain the projection plane and projection direction with the least data loss after projection of 3D data according to the maximum variance theory.

Given the point cloud matrix $P$, conventional PCA finds the optimal $r$ rank matrix $L$, which can be solved by SVD decomposition of the eigenvalues. For noisy point cloud data, the solved result will be very inaccurate. RPCA decomposes $P$ into two parts $P = L + S$, where $L$ is the low-rank part and $S$ is the sparse part, and the optimization objective of RPCA can be expressed by the following equation:

$$\min_{L,S} \text{rank}(L) + \lambda \|S\|_0 \\ s.t. \quad P = L + S \tag{10}$$

Since the above equation has nonconvex and nonsmoothed properties in optimization, it can be transformed into a relaxed convex optimization problem, and the optimization of RPCA can be performed by methods such as the augmented Lagrange multiplier method.

$$\min_{L,S} \|L\|_* + \lambda \|S\|_1 \\ s.t. \quad P = L + S \tag{11}$$

*3.2. Large-Scale Noise Removal*

For large-scale noise, the first type of noise is generally removed by the enclosing box method and the radius filter method, and the second type of noise is removed by the region growth method and the statistical filter method, which can effectively remove the first and second types of noise, respectively, but different methods are needed to remove different types of noise in this way, and the efficiency is low. Therefore, a median denoising method based on region segmentation is used.

The set of point cloud coordinates transformed by the coordinate system is $Q$. The point cloud region is segmented along the $u_1$ direction, and the set of $u$ coordinates of data points in each region is $Q_j^u[z'', \sigma](j = 1, 2 \cdots, q)$, $\sigma$ denotes the $u_1$ direction segmented point cloud region. The $\sigma$ values are set according to the laser scanning frequency of the acquisition system; as shown in the figure, $Q_j^u[z_k'', \sigma](j = 1, 2 \cdots, q)$ is the set of $u$ coordinates in the segmented area. $Q_j^u[z'', \sigma](j = 1, 2 \cdots, q)$ can be expressed as:

$$Q_j^u[z'', \sigma] = \left\{ z_k'' \,\middle|\, x_j'' \leq x_k'' \leq x_j'' + \sigma \right\} \tag{12}$$

where $z_k''(k = 1, 2 \cdots, l)$ denotes the $u$ coordinates of the data points in the region; $l$ denotes the number of $u$ coordinates of the data points in the region; $x_j''$ and $x_j'' + \sigma$ are the $u_1$ coordinates of the $j_{th}$ segmentation region boundary.

The elements of $Q_j^u[z_k'', \sigma](j = 1, 2 \cdots, q)$ are sorted in the segmentation region, the median $z_m''$ of the elements in the segmentation region is obtained, the distance between each element of $Q_j^u[z_k'', \sigma](j = 1, 2 \cdots, q)$ and $z_m''$ is calculated and recorded as $d_k = |z_k'' - z_m''|$, $d_k$ is arranged in the order from smallest to largest, and the threshold $d_m$ is set, remove $z_k''$ for $d_k$ values greater than the threshold $d_m$, and mark the remaining set of elements corresponding to the point cloud data sitting as $Q'$.

*3.3. Small-Scale Noise Removal*

For the filtering of small-scale noise, traditional filtering methods are used such as mean filter [24], median filter [25], Gaussian filter [26], and bilateral filter [27]. The essence of mean filtering is to distribute the outliers around each point equally to all points, and point clouds are irregularly distributed in most cases, which causes mean filtering to potentially degrade accuracy when denoising point clouds. Median filtering method has the drawback of losing details and connectivity on the point cloud surface, as it is hard to determine the optimal size of the neighborhood for denoising. Gaussian filtering method blurs the edges more obviously, and the protection of high frequency details is not obvious. The bilateral filtering method needs to use the K-neighborhood in the process of filtering and solves the bilateral filtering factor of all data points in the neighborhood at that point, which is too long in computation time and low in efficiency and cannot be well applied to field experiments. Therefore, a fast bilateral filtering algorithm based on threshold segmentation is proposed in this paper. The specific algorithm is as follows.

Firstly, the point cloud data are divided into grids along $u_1$ axis and $u_2$ axis direction, respectively, and the set of $u$ coordinates in the grid is $\widetilde{Q}^u[z'', \lambda]$, $\lambda$ denotes the length of the divided grid, and $\widetilde{Q}^u[z'', \lambda]$ is expressed as

$$\widetilde{Q}^u[z'', \lambda] = \left\{ z_k'' \,\middle|\, \begin{matrix} x'' \leq x_k'' \leq x'' + \lambda \\ y'' \leq y_k'' \leq y'' + \lambda \end{matrix} \right\} \tag{13}$$

where $z_k''(k = 1, 2 \cdots, s)$ represents the $u$ coordinates of the data points in the grid; $s$ represents the number of $u$ coordinates of the data points in the region; $x''$ and $y''$ are the grid boundary coordinates. Before the bilateral filtering, the points in the grid are thresholded for segmentation to find the outliers in the grid.

Sort the elements of $\widetilde{Q}^u[z'', \lambda]$ in the grid, find the median $\widetilde{z}_m''$ of the elements in the grid, calculate the distance between each element of $\widetilde{Q}^u[z'', \lambda]$ and $\widetilde{z}_m''$ as $\widetilde{d}_k = |z_k'' - \widetilde{z}_m''|$, arrange $\widetilde{d}_k$ in the order from smallest to largest, and set the threshold $\widetilde{d}_m$, $z_k''$ whose $\widetilde{d}_k$ values greater than the threshold $\widetilde{d}_m$ are median filtered, and the 3D data corresponding to the elements that meet the requirements are used as neighborhood points to calculate the smooth filtering power function and feature retention power function, and to obtain the new bilateral filtering factor.

To improve the operational efficiency, the improved bilateral filtering factor can be expressed as:

$$\alpha' = \frac{\sum\limits_{j=1}^{N} w_c(\|q_i - q_j\|) w_s(\|z_i - z_j\|)(z_i - z_j)}{\sum\limits_{j=1}^{N} w_c(\|q_i - q_j\|) w_s(\|z_i - z_j\|)} \tag{14}$$

where $N$ is the number of neighboring points, $w_c(x) = e^{-x^2/(2\sigma_c^2)}$ is defined as the smooth filtering power function, $w_s(x) = e^{-x^2/(2\sigma_s^2)}$ is the feature retention power function, $\sigma_c$ and $\sigma_s$ are the Gaussian filtering factors, $\sigma_c$ is the length of the grid, $\sigma_s$ is the length of the points in the grid, and $q$ is the standard deviation of the distance from the point in the grid to the point.

The filtered point cloud data are:

$$q' = q - \alpha' n \tag{15}$$

where $q$ is the pre-filtered point cloud data, $q'$ is the post-filtered point cloud data, and $n$ is the unit vector (0,0,1).

## 4. Experimental Validation and Analysis

The validation of the proposed method is carried out using the dataset and the effectiveness of the proposed method is verified in the process of large aircraft assembly.

### 4.1. Point Cloud Resampling Method

The validity of the method is verified using the public dataset from Stanford University [28]. The Stanford Bunny is a 3D test model widely used in the field of computer graphics. The model contains data on 69,451 triangles that were determined by a 3D scanner to form a ceramic rabbit statue. First, the original point cloud data are transformed and the transformed point cloud data are resampled, and then the effectiveness of the proposed method is verified by comparing the alignment accuracy of the resampled data with the original point cloud (Figure 2).

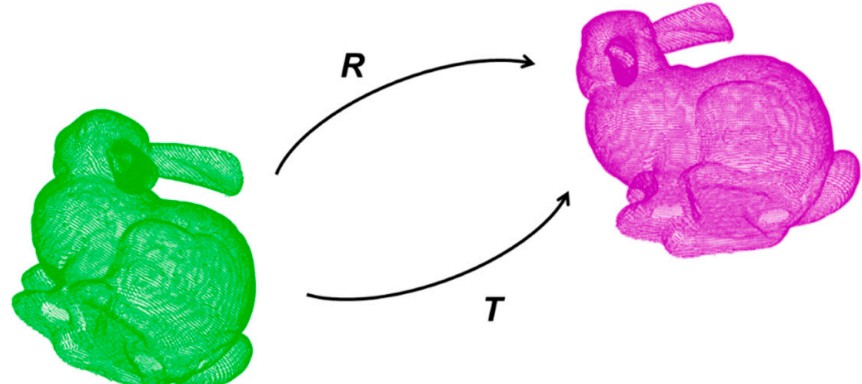

**Figure 2.** Three-dimensional spatial transformation process of point cloud data by rotation matrix R and translation matrix T.

The point cloud resampling results are shown in Figure 3. From the figure, the other sampling methods cannot achieve the balance between point cloud feature retention and point cloud uniformity, which can easily lead to missing point cloud features (e.g., points at the rabbit's foot profile). In addition, the proposed method has adjustable hyperparameters that can be easily defined in conjunction with point cloud data processing requirements.

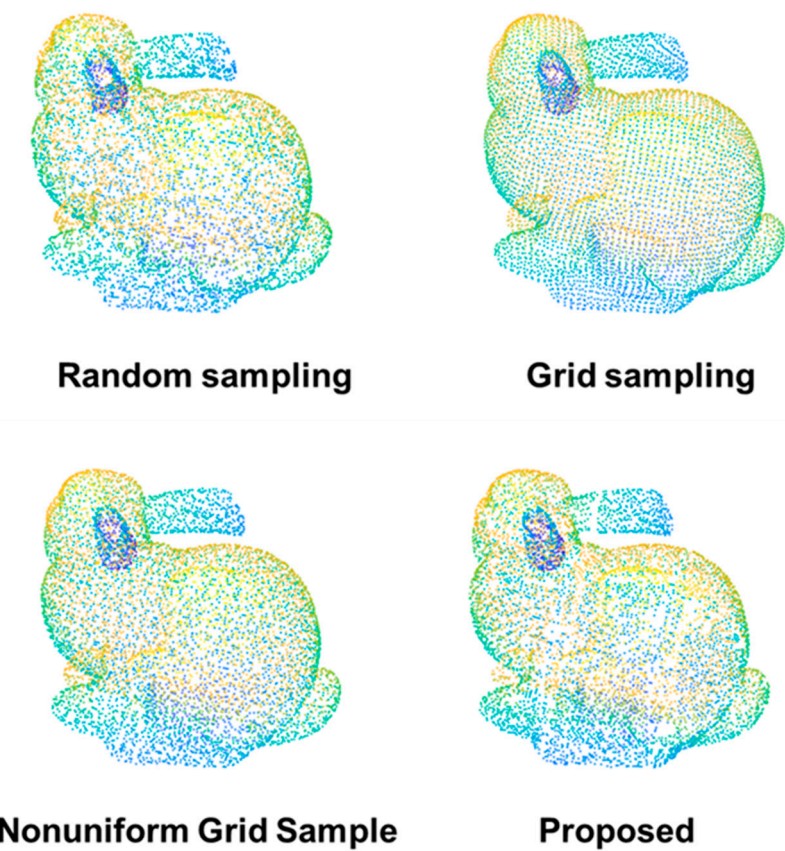

**Figure 3.** Point cloud data resampling comparison results of different methods.

To compare alignment errors, the point cloud is rotated and translated in the space. The results of point cloud alignment using the Iterative Closest Point (ICP) algorithm are shown in Figure 4. There are obvious alignment deviations in other methods, while the proposed method is basically deviation-free. The misalignment angles and translations between the aligned point cloud and the original point cloud are evaluated, and the results are shown in Table 1. The $\Delta\theta_x$, $\Delta\theta_y$, and $\Delta\theta_z$ are the misalignment angles of the *x*-axis, *y*-axis, and *z*-axis, respectively. The $\Delta T_x$, $\Delta T_y$, and $\Delta T_z$ are the misalignment translations of the *x*-axis, *y*-axis, and *z*-axis, respectively. As can be seen from the table, the proposed method outperforms the other methods in almost all alignment errors. Among other methods, the nonuniform method has the lowest error. Compared with the nonuniform method, the proposed method reduces the maximum misalignment angle deviation from 0.0086 rad to 0.0041 rad in absolute value and reduces the maximum misalignment translation deviation from 3.12 mm to 0.65 mm.

*4.2. Point Cloud Denoising Method*

In order to verify the proposed point cloud noise filtering method, a 10% proportion of Gaussian noise and random noise are added to the dataset and the results of different methods are compared. The proposed method was compared with the commonly used Gaussian filter, mean filter, and median filter methods, and the comparison results are shown in Figure 5. From the figure, it can be seen that Gaussian filtering is less effective, and mean filtering and median filtering can filter out part of the noise, while the proposed method basically achieves effective filtering of all the noise points.

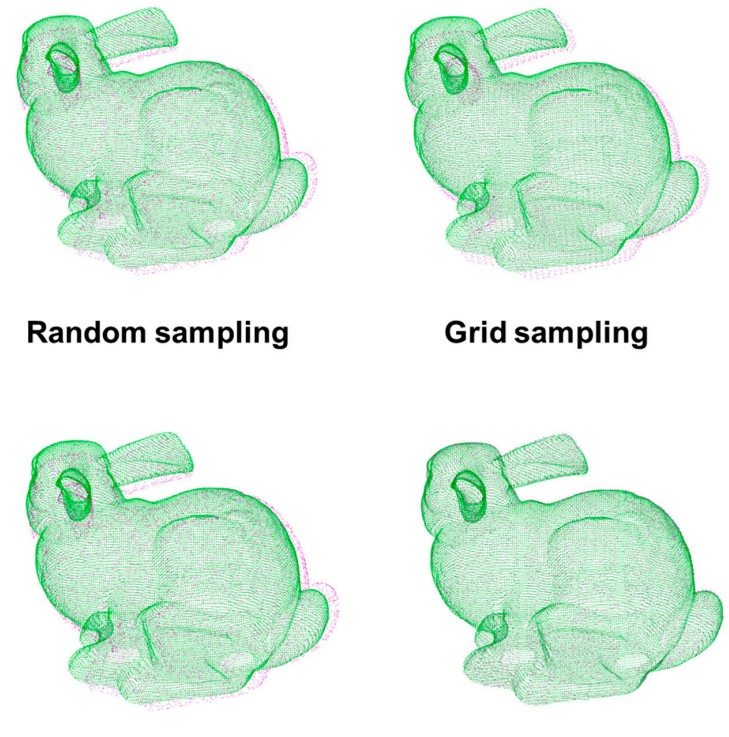

**Figure 4.** Comparison of the alignment accuracy of different resampling methods.

**Table 1.** Comparison of the alignment accuracy.

|  | $\Delta\theta_x$ (rad) | $\Delta\theta_y$ (rad) | $\Delta\theta_z$ (rad) | $\Delta T_x$ (mm) | $\Delta T_y$ (mm) | $\Delta T_z$ (mm) |
|---|---|---|---|---|---|---|
| Random | $-0.0039$ | 0.0088 | $-0.0075$ | $-3.20$ | 0.49 | 4.71 |
| Grid | $-0.0034$ | 0.0108 | $-0.0070$ | 3.21 | 0.34 | 5.13 |
| Nonuniform | 0.0030 | 0.0086 | $-0.0015$ | 1.32 | 0.77 | 3.12 |
| Proposed | $-0.0018$ | $-0.0023$ | $-0.0041$ | $-0.25$ | $-0.25$ | 0.65 |

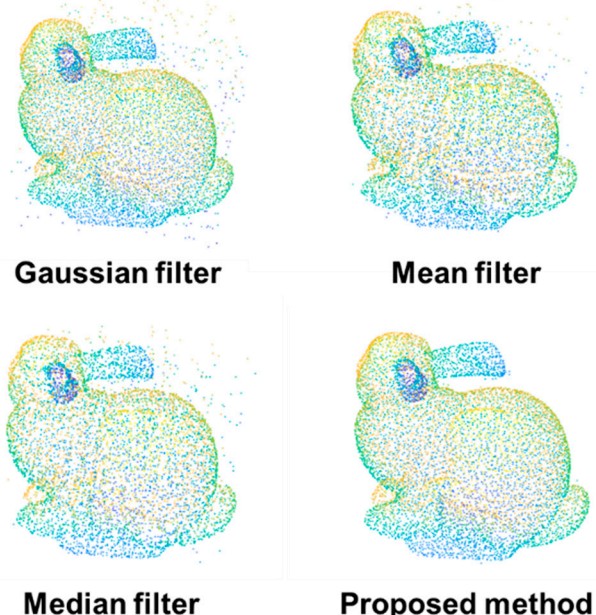

**Figure 5.** Comparison of point cloud denoising methods.

In addition, a quantitative analysis of the effect of noise filtering was performed. Different proportions of noise were added to the dataset and the noise filtering rate was analyzed, and the results are shown in Figure 6. Noise percentage refers to the percentage of noise points to the total number of points in the point cloud. It can be seen that the filtering rate gradually decreases as the proportion of noise increases, but all of them exceed 84%. That is, more than 84% of the added noisy data points are effectively removed. This indicates that the proposed method has good robustness and can guarantee the effective noise filtering of the point cloud dataset.

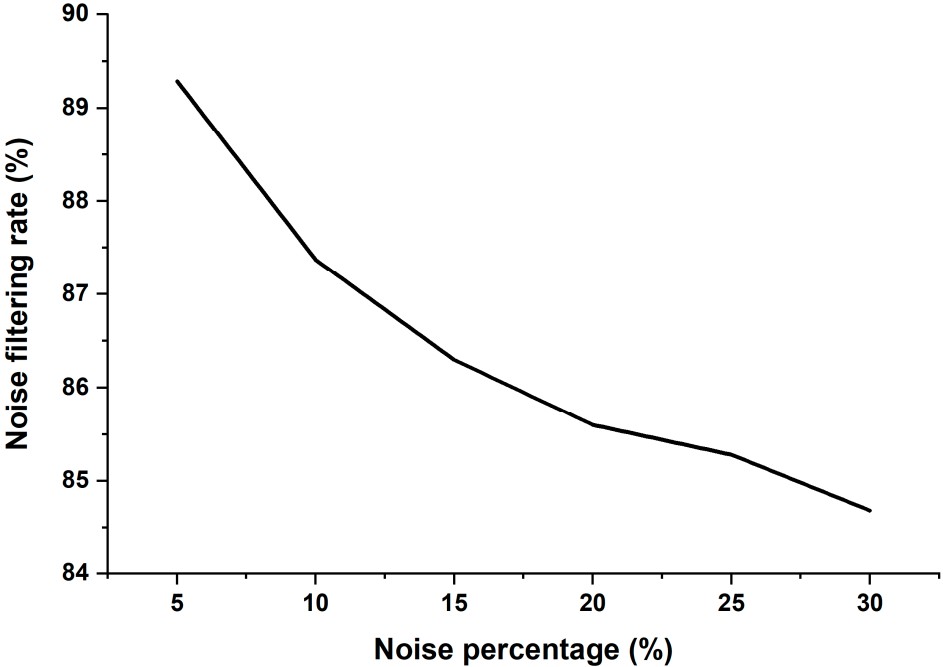

**Figure 6.** Analysis of noise filtering result by adding different percentages of noise.

### 4.3. Experimental Verification

The effectiveness of the proposed method was verified in the aircraft wing-body assembly process, as shown in Figure 7. The left wing box of the aircraft was scanned using a handheld scanning device, and the point cloud data of the wing box was obtained, as shown in Figure 8. The original point cloud data contain about 1.04 million data points and contain a lot of noise, and are difficult to analyze and align.

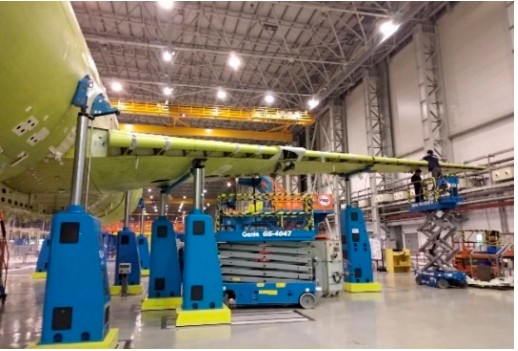 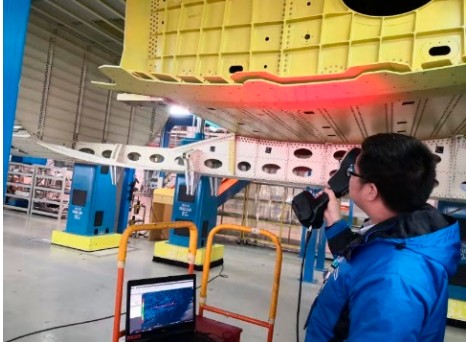

**Figure 7.** Measurement experiments during wing-body assembly.

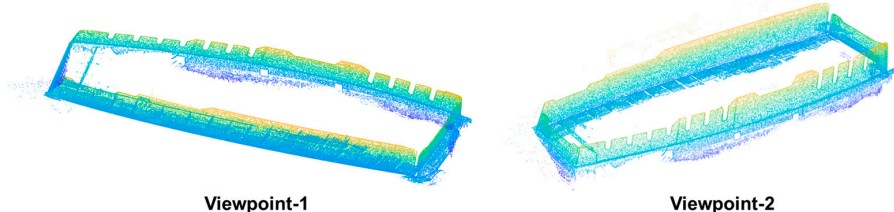

**Figure 8.** Original point cloud data of the wing box.

The proposed method was used to analyze the wing box point cloud data. The point cloud resampling results are shown in Figure 9, and the point cloud denoising results are shown in Figure 10. In the process of point cloud resampling, the data volume is reduced to 5% of the original, and it can be seen from the figure that the proposed method retains the edge features of the point cloud well, and the denoising results after resampling also show that the proposed method eliminates the point cloud noise.

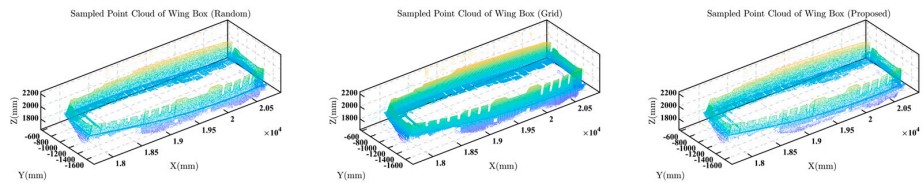

**Figure 9.** The point cloud resampling results of the wing box.

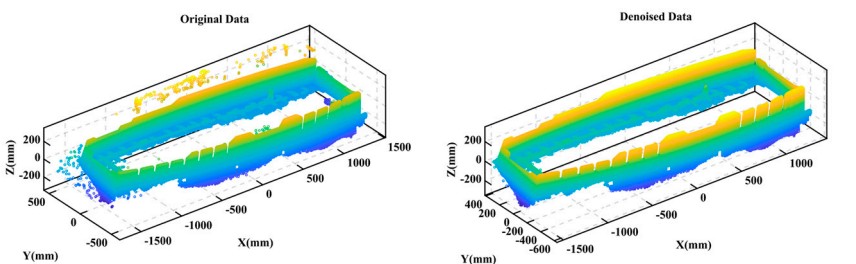

**Figure 10.** The point cloud denoising result of the wing box.

In terms of analysis time, the time cost is extremely high for the analysis of the original point cloud. The analysis of point cloud data by a workstation equipped with i7-10700K and 32 GB memory takes more than 3 h to align, but the overall time of the proposed method is less than 5 min, which can greatly improve the efficiency of aircraft assembly.

Furthermore, as shown in Figure 11, we align the wing point cloud data with the theoretical numerical model to obtain the deviation between the wing box area and the numerical model, which provides a data basis for the evaluation and prediction of the gap between the wing and the central wing box.

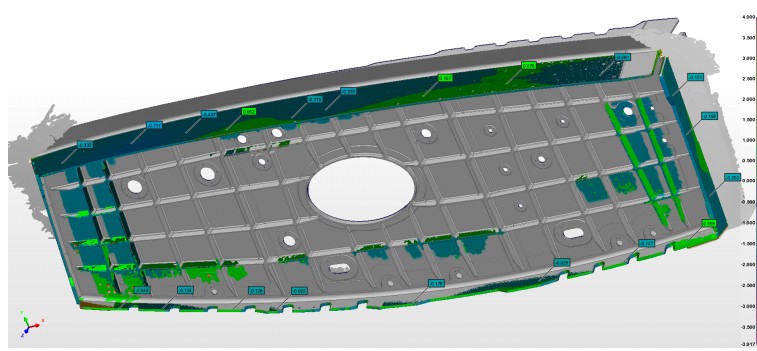

**Figure 11.** The result of wing box point cloud data alignment and gap evaluation.

## 5. Conclusions

This paper investigates the problem of pre-processing point cloud data in aircraft assembly. To solve the problem of the high cost of the alignment and analysis time due to redundant point cloud data, the nonuniform voxel mesh resampling and graph signal processing methods are combined. Validation results on a standard dataset show that the proposed method reduces the maximum misalignment angle deviation from 0.0086 rad to 0.0041 rad in absolute value and reduces the maximum misalignment translation deviation from 3.12 mm to 0.65 mm compared with the nonuniform method. For the noise of the point cloud, a multi-scale point cloud noise filtering method is proposed. The method can remove both the large-scale noise and small-scale noise effectively. Finally, the proposed method was validated in the process of large passenger aircraft wing-body assembly. The analysis of the left wing box point cloud data containing 1.04 million data points collected with a handheld scanning device showed that the proposed method reduced the processing time of the point cloud data from three hours to less than five minutes, effectively guaranteeing the efficiency of aircraft assembly. Subsequently, the robustness of the algorithm needs to be further improved and the deviation of the point cloud data from the numerical model needs to be analyzed, so as to provide a reference for the development of the regulation algorithm in the aircraft assembly process.

**Author Contributions:** Conceptualization, W.H.; Methodology, R.L. and W.H.; Software, R.L. and S.L.; Validation, S.L.; Formal analysis, R.L. and W.H.; Investigation, S.L.; Resources, S.L.; Data curation, S.L.; Writing—original draft, R.L. and W.H.; Writing—review & editing, R.L. and S.L.; Visualization, R.L. and W.H.; Supervision, S.L.; Project administration, S.L.; Funding acquisition, S.L. All authors have read and agreed to the published version of the manuscript."

**Funding:** This research was funded by [Science and Technology Innovation Action Plan of Shanghai] grant number [21511102600].

**Conflicts of Interest:** The authors declare no conflict of interest.

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
