# Peer review of "Three-Dimensional Point Cloud Data Pre-Processing for the Multi-Source Information Fusion in Aircraft Assembly"

_applsci, doi:10.3390/app13084719_

Round 1

Reviewer 1 Report

The manuscript has a relatively acceptable flow and structure overall. It also seems to have an interesting topic for the readers; however, there are a number of issues throughout the entire manuscript that need to be fixed. The following points are the most significant issues that this reviewer recommends the authors to consider for enhancing the quality and writing of the manuscript:

The authors may need to provide justifications for using data processing and information fusion techniques for the purpose of this aircraft assembly.

The number of references may not seem sufficient for a journal. The authors may need to use and refer to the most recent publications on the topic of this manuscript. Although there are some of the very recent publications among the references, it seems not to be sufficient.  

The authors should further emphasize on the novelty of their research by providing additional recent references and making pair-wise comparison to prove their contribution to the body of knowledge.

The mathematical equations and formulas in Sections 2 and 3 need further elaborations; many of the provided equations and numerical formulas require additional explanation. The current way the authors developed this sections is ambiguous especially for the readers who are not fully familiar with the concepts of information fusion in aircraft assembly.

Regarding Figure 8, the authors are encouraged to show it from other viewpoints. The current provided figure is only a top-view.

The authors may need to further mention the limitations of their study in the conclusion section of the manuscript.

The conclusion section is very short and limited. The authors may need to provide a more comprehensive conclusion.

Reviewer 2 Report

This paper has been very poorly prepared. Although the topic is worthy of publication, the paper itself requires major revision. Some of the problems:

1) The English needs attention, and in places there are incomplete sentences. Moreover there are two instances of repeated sections of text, one a sentence (top of pg 8) & the other a full paragraph (top of page 9).

2) A "Fig X" is referred to (page 3), but it does not appear to exist.

3) Some references are missing, eg to Fleishman on pg. 6

4) In Section 4 there's first reference to "using the data set". What data set? There's been no information on any data set up to this point.

5) Table 1 is said to list alignment accuracy, but there's no indication of just what the listed variables are, nor their units and the presumed (??) transformation parameters Ri & Ti are not defined.

6) The figure caption for Fig. 4 reads as Fig. 3, and the figures in Figs. 3 - 5 are too small to visually interpret adequately; the resolution is especially bad when the paper is printed.

7) There is a strange sentence below the caption for Fig. 6 which starts "This section is not mandatory but can be added to the manuscript ..." 

8) Figure captions are in most case insufficiently descriptive of te figure contents.

9) The explanation of the filtering rate versus noise percentages needs more explanation. Just what is "84%"?

10) The experimental verification section is far too cursory and more details and analysis are warranted.

11) In the conclusions, it states that "the method reduces the alignment deviation from 6.92% to 0.86%", but exactly what the percentages refer to (eg angular misalignment in a given coordinate axis?) is not explained.

In short the paper needs a comprehensive revision.  

Round 2

Reviewer 1 Report

Thank you for addressing my comments. I do not have any further comments.

Author Response

Thanks for your valuable suggestions and hard work.

Reviewer 2 Report

1) The English needs carefully checking. Errors highlighted in the first review have not been fixed - they need to be! Examples of poor sentences, just a few are as follows:

" Chen et al [19] proposed a resampling strategy based on graph filter for the storage and processing of large scale point cloud for the difficulty. Paoli et al [20] ..."

"Given the point cloud matrix P, conventional PCA finds the optimal r rank matrix L, which makes, can be solved ..."

"For the filtering of small-scale noise, traditional filtering methods such as mean filter [24], median filter [25] , Gaussian filter [26] and bilateral filtering[27]. The ..." Sentence needs completing.

" Median filtering is difficult to determine the size of the neighborhood when denoising, which tends ..."

" ...the points in the grid are thresholder for segmentation ..."

2) A major problem is Table 1. As mentioned in the first review, the terms Ri & Ti are not defined and there are no units. Hence the table is meaningless to the reader. It is not good enough to say that R & T are parameters of  rotation and translation matrixces, respectively. What parameters? A seven parameter transformation has 3 rotations, a scale factor & three translations. The Ti terms might (??) correspond to DX,DY,DZ, but we don't know, and what are the units? And what of Ri? They cannot be rotation angles because there would only be 3. And, if they are matrix elements, why are there 4 & not 9? Listing the 9 is anyway as useless as listing 4 that are undefined. The aim here is to list misalignment accuracy, so surely the best terms to list would be angular misalignment, with units so the reader can physically interpret the values. As mentioned, right now the table tells the reader nothing because he/she does not know what these values are. This MUST be fixed!

Round 3

Reviewer 2 Report

Inclusion of the transformation about the Z axis, and hence Ri & Ti is helpful, but ...

The authors still need to Fix up Table 1. There are still no units for Ti & the parameters Ri have no meaning to a reader. Why not simply list the misalignment angle ThetaZ, from which Ri's are calculated. An angle has meaning to a reader; matrix element differences are not interpretable in a physical sense. Why complicate things? List the misalignment angle, ThetaZ & the Ti's AND their units. 
